# Cold Sulfur Springs—Neglected Niche for Autotrophic Sulfur-Oxidizing Bacteria

**DOI:** 10.3390/microorganisms11061436

**Published:** 2023-05-29

**Authors:** Lea Nosalova, Maria Piknova, Mariana Kolesarova, Peter Pristas

**Affiliations:** 1Department of Microbiology, Faculty of Science, Institute of Biology and Ecology, Pavol Jozef Safarik University in Kosice, 041 54 Kosice, Slovakia; lea.nosalova@student.upjs.sk (L.N.); maria.piknova@upjs.sk (M.P.); mariana.kolesarova@upjs.sk (M.K.); 2Centre of Biosciences, Institute of Animal Physiology, Slovak Academy of Sciences, 040 01 Kosice, Slovakia

**Keywords:** sulfur-oxidizing bacteria, cold sulfur springs, ecology, freshwater, sulfur

## Abstract

Since the beginning of unicellular life, dissimilation reactions of autotrophic sulfur bacteria have been a crucial part of the biogeochemical sulfur cycle on Earth. A wide range of sulfur oxidation states is reflected in the diversity of metabolic pathways used by sulfur-oxidizing bacteria. This metabolically and phylogenetically diverse group of microorganisms inhabits a variety of environments, including extreme environments. Although they have been of interest to microbiologists for more than 150 years, meso- and psychrophilic chemolithoautotrophic sulfur-oxidizing microbiota are less studied compared to the microbiota of hot springs. Several recent studies suggested that cold sulfur waters harbor unique, yet not described, bacterial taxa.

## 1. Introduction

The biogeochemical sulfur cycle plays an important role in the regulation of Earth’s conditions and virtually in Earth’s evolution. It is a complex web of biotic and abiotic sulfur transformations mediated by spontaneous chemical reactions and specialized forms of life. From the dawn of unicellular life, the chemolithoautotrophic metabolic activities of bacteria have been an important part of the sulfur cycling due to their enormous capacity for redox transformations of inorganic compounds. As they are able to transform inorganic substrates into organic molecules, they represent secondary, and in some environments primary, producers of biomass [1,2,3,4,5]. The oxidation of reduced inorganic sulfur compounds is driven by an extensive range of phototrophic and chemolithotrophic sulfur-oxidizing microorganisms occupying diverse, mostly extreme or moderate environments and having evolved different pathways for sulfur oxidation [6]. Although sulfur oxidation and sulfur reduction processes have received the attention of microbiologists, our knowledge of the metabolic pathways and bacteria using them is limited. In recent years, completely new lineages of sulfur-oxidizing and -reducing bacteria were described; nevertheless, numerous microorganisms remain unexplored [7,8]. Cold sulfur springs represent neglected niches for sulfur-oxidizing bacteria, and recent studies suggested that they harbor unique, non-explored sulfur-oxidizing bacterial communities.

## 2. Main Body

### 2.1. The Evolution of Sulfur-Oxidizing Bacteria Was Coupled with the Evolution of Earth

Sulfur bacteria have been present on Earth since the beginning of life and are of vital importance for the current functions of ecosystems [9]. The history of Earth’s evolution coupled with the evolution of prokaryotic life is an example of a biological–geological loop, where biologically driven processes influenced the biochemistry of Earth [10]. Chemolithoautotrophic life based on sulfur sources is thought to have originated from ecosystems present on early Earth and probably enabled the evolution of life based on energy sources other than sulfur [6]. One can imagine an early Earth ecosystem as a terrestrial environment surrounding active hydrothermal areas with primary organic matter and oxidized sulfur species produced by anoxygenic photosynthesis [11]. Correspondingly, sulfur isotope analyses of microfossils from rocks from the Archean era suggested that life with sulfur-based metabolism existed almost 3.5 billion years ago; therefore, it represents one of the first forms of life [12,13]. From the late Archean to the early Proterozoic era, environments had a reduced anoxygenic or low oxygen atmosphere, and sulfide and sulfur served as electron donors for photosynthetic carbon fixation. The reduction of sulfate and sulfur was probably the main respiratory process in that era [14]. There is evidence of sulfate reduction coupled with sulfur disproportionation around 3.5 billion years ago, and the reduction of oxidized sulfur has remained an important metabolic pathway in anoxygenic environments [15]. Anoxygenic photosynthesis has provided sulfate for sulfur-reducing bacteria, and with oxygenic photosynthesis, non-photosynthetic sulfide oxidation appeared obviously, and the evolution of microorganisms was associated with innovations in metabolic pathways [11].

The rock records of sulfur and carbon isotopes provide convincing evidence of the transition from an anoxygenic atmosphere to an atmosphere with free oxygen, and phylogenetic and comparative analyses may shed light on the early stages of microbial evolution [16,17]. The Great Oxidation Event (2.4–2.2 billion years ago) denoted the time of oxygen concentration increasing and simultaneously represented the increase in the production of metabolically unstable sulfate, which mutually had a crucial impact on the sulfur cycle [18]. Changes in the sulfur cycle resulted in changes in surface, atmosphere, and ocean chemistry and were accompanied by changes in sulfur-oxidizing bacteria ecology [19]. It is suggested that the evolutionary radiation of microorganisms is promoted by severe environmental changes rather than slow successive changes in environmental parameters. One of the major changes in environmental factors is the availability of oxygen, which may have been the driving force of non-photosynthetic sulfur-oxidizing bacteria evolution [11].

According to fossil records, sulfur-oxidizing bacteria were present on Earth during the Archean era and the Great Oxidation Event [20], and an increase in oxygen concentration instigated the evolution of bacteria and led to changes in the bacterial ecology [11]. Prokaryotic organisms have evolved a plethora of metabolic pathways encoded by genes to gain energy through redox sulfur reactions [10,21]. Presumably, sulfur oxidation in the energy generating process developed from different metabolic pathways. One example may be Dsr (dissimilatory sulfite reductase), whose homologs were found in methanotrophs inhabiting sulfur-rich hydrothermal environments; the enzyme has an ability to transform sulfite to sulfide. As the bacteria began to colonize new microhabitats with a high concentration of sulfite, they evolved a metabolic pathway to detoxify it. The competing back reaction might be a reaction with sulfur, producing hydrogen sulfide, thiosulfate, and/or polysulfides [22,23]. After the evolution of the first pathways of sulfur respiration, other metabolic pathways followed using other reduced sulfur sources. Sulfur is an intermediate in reduced sulfur species oxidation, however, during the Eoarchean–Paleoarchean the absence of isotopic evidence for sulfur oxidation to sulfate may suggest that oxidation of sulfur originated in the Neoarchean era [23]. Phylogenetic studies based on *dsr* and *aps* gene sequences showed the ancient nature of sulfate/sulfur reduction coupled with sulfide oxidation pathways [24,25,26]. During the Protozoic era, a trend of increasing oxygen levels continued, which resulted in the decreasing level of sulfide with the opposite effect on less reduced sulfur species. This event influenced the diversification of sulfur-oxidizing bacteria [26,27]. During the Proterozoic era, i.e., 0.75–0.62 billion years ago, more sophisticated metabolic pathways for sulfur oxidation evolved [26]. According to Meyer et al. [28], the *sox* phylogenetic analysis indicated several events of lateral gene transfer among sulfur-oxidizing bacteria, e.g., the phototrophic sulfur-oxidizing bacteria acquired the *soxB* gene from their chemolithotrophic donors. However, nowadays there are conspicuous differences in the *sox* gene cluster between phototrophic and chemotrophic bacteria as evolutionary changes continued [29].

### 2.2. Sulfur Supporting Life

Sulfur is an essential element that possesses various oxidation states from −2 to +6 [2], of which three are abundant in the environment; −2, as in sulfide; 0, as elemental sulfur; and +6, as in sulfate [30]. Sulfate is the stable form of sulfur in the presence of oxygen, which is abundant mostly in the marine environment, whereas stable reduced sulfur sources are sulfides and elemental sulfur, which are relatively common in anoxic environments [5]. These compounds are continuously transformed into each other by chemical, biological, and geological processes. The transformation of inorganic sulfur compounds, as well as organic compounds, is, to a limited degree, carried out by microorganisms [30]. Summarily, the biogeochemical sulfur cycle is a description of reactions that transfer sulfur through different reaction states and ecosystems and encompasses abiotic and biologically mediated reactions [8,31]. Furthermore, the sulfur cycle is linked to other essential cycles, including nitrogen, carbon, and metals via metal sulfides [32]. Similar to other essential biogeochemical cycles, the sulfur cycle also has oxidative and reductive sites, and microorganisms perform an irreplaceable function in balancing them. Imbalances of the sulfur cycle reactions lead to the accumulation of intermediates in the environment [33]. Bacteria are also capable of sulfur, thiosulfate, and sulfite disproportionation [34]. In contrast to sulfur assimilation for sulfur-containing cell constituents, which is present in most taxonomic groups, sulfur dissimilation is restricted to prokaryotes as an energy yielding process with sulfur ions as donors or acceptors of electrons [35].

The transformation of organic sulfur compounds to inorganic ones is performed by anaerobic sulfur bacteria. For example, up to 50% of the organic matter in marine sediments is mineralized by sulfur-reducing bacteria. This process of organic matter mineralization is linked to the food web of organic degradation and connects the sulfur and carbon cycles [15,36,37]. In light-independent environments, the primary biomass production is driven by chemolithoautotrophy. In sulfur-rich environments, sulfur cycling is an important driver of microbial growth and element conversion in many ecosystems [7,32,38]. Biological sulfur oxidation is a vital part of the sulfur cycle, and in natural environments the oxidation of sulfur species is mediated by chemolithotrophic and phototrophic bacteria and by representatives of the order *Sulfolobales* of the domain *Archaea* [39,40]. Luther et al. [22] stated that bacterial enzymes evolved to overcome chemical oxidations, and, in most environments, both anaerobic photolithotrophic and aerobic chemolithotrophic sulfide oxidation rates were three or more orders of magnitude higher than abiotic rates. The biological and chemical oxidation of sulfide produced has established a complex network of redox sulfur processes. The intermediates of sulfur oxidation are represented by sulfite and polysulfide sulfur thiosulfate, which are all substrates for sulfur-oxidizing bacteria [37]. The diversity of sulfur redox states has given rise to a variety of enzymes and metabolic pathways that are used to transform sulfur species. Among many transformation processes of the sulfur cycle, some of them are used by prokaryotic organisms to generate reductants and ATPs from inorganic sulfur compounds [26]. For photosynthetic bacteria, sulfur species represent a source of electrons for CO_2_ reduction under anoxic autotrophic growth, whereas chemolithotrophic bacteria use derived electrons for CO_2_ fixation and as respiratory electron donors [41]. New discoveries of sulfur bacteria and metabolic pathways have demonstrated the complexity of the biological sulfur cycle [37,42,43,44].

### 2.3. Sulfur-Oxidizing Bacteria Riding the Biogeochemical Sulfur Cycle Using Different Metabolic Pathways

Reduced sulfur species using microorganisms as energy sources gained in the process of sulfur oxidation include sulfides (S^2−^, HS^−^, H_2_S), polysulfides (^−^S-S_n_-S^−^), elemental sulfur (S^0^), tetrathionate (S_4_$O_{6}^{2-}$), sulfite (S$O_{3}^{2-}$), and thiosulfate (S_2_$O_{3}^{2-}$) [45]. The diversity of sulfur-oxidizing bacteria is reflected in the diversity of sulfur redox states [46]. A variety of sulfur-oxidizing bacteria use sulfur as an energy source, and this group encompasses bacteria with little or no taxonomic relationship to each other [47]. They have evolved a variety of seemingly redundant enzymatic pathways of sulfur transformation [28,48]. Aerobic sulfur-oxidizing bacteria, so-called colorless sulfur-oxidizing bacteria, are mainly mesophilic, and it was believed that most of them belong to the phylum Pseudomonadota [36,49]. However, several studies have suggested that the phylogeny of chemolithoautotrophic sulfur-oxidizing bacteria is more complex than previously expected [50,51,52,53,54]. Phototrophic sulfur-oxidizing bacteria are phylogenetically diverse, mostly neutrophilic and mesophilic, and are divided into two groups, green sulfur-oxidizing and purple sulfur-oxidizing bacteria with representatives, e.g., *Chlorobiaceae*, *Chromatiaceae*, and *Rhodobacteraceae* [39,55]. Additionally, several representatives of the domain *Archaea* are also considered to be sulfur-oxidizing.

Representatives of the domain *Archaea* constitute a considerable part of the microbial Earth’s biomass and play a crucial role in the Earth’s global geochemical cycles, including the sulfur cycle. Sulfur-dependent archaea are mostly found in hydrothermal environments [56]. Considering the diversity of sulfur oxidation states, representatives of the domain *Archaea*, order *Sulfolobales*, and *Thermoplasmatales,* respectively, have evolved a variety of enzymes engaged in sulfur dissimilatory metabolism. In the order *Sulfolobales*, many enzymes and proteins of sulfur metabolism exist, including sulfur-reducing enzymes, sulfur-oxidizing enzymes, sulfur transferases, and sulfur carrier proteins [45,56]. Some of these processes are shared with bacteria, however, some of them are unique [57]. The model organism for sulfur oxidation in this group is obligate chemolithotrophic *Acidianus ambivalensis*, which oxidizes elemental sulfur resulting in sulfuric acid production. The oxidation consists of two steps; first, sulfur is oxidized to sulfite by sulfur oxygenase reductase (SOR) or sulfur dehydrogenase, and then, by the activity of sulfite:acceptor oxidoreductases or dehydrogenases sulfite is transformed to sulfuric acid. During the process, other intermediates are produced, such as thiosulfate, tetrathionate, and others, which are presumably subsequently oxidized during the energy conservation process [58]. Thiosulfate oxidation was characterized at the molecular level. Thiosulfate:quinone oxidoreductase (TQO) oxidizes thiosulfate, producing tetrathionate. The TQO complex consists of two subunits, DoxA and DoxD, and homologs of their genes are also present in other *Crenarchaeota* [57]. Tetrathionate is unstable in the presence of strong reductants and, thus, is probably re-reduced with H_2_S produced by the SOR, indirectly feeding the quinone pool by electrons from S^0^ disproportionation. In this process, caldariella quinone serves as the electron acceptor and the terminal aa3-type quinol oxidase shuttles electrons from the caldariella quinone pool to O_2_ [57,59]. Moreover, the tetrathionate hydrolase (TTH) activity was observed in cultures of *A. ambivalens* grown with tetrathionate as the sole sulfur source. Contrary to that, the activity of this enzyme was not observed in cultures with other sulfur sources, and homologs of *tth* genes were not found in the genomes of other *Archaea* [60]. Lastly, our knowledge about sulfur oxidation in *Archaea* is not complete as many pathways have not yet been discovered. 

Thiosulfate is thought to fulfill the key role in the biogeochemical sulfur cycle as it is an energy source of most sulfur-oxidizing bacteria. Besides the dissimilatory sulfur pathway of representatives of the domain *Archaea*, there are currently at least three relatively well-described pathways that are postulated to be used by representatives of the domain *Bacteria* [6,28,38]. These include the complete oxidation of reduced sulfur compounds with the end product of sulfate, known as the Sox metabolic pathway, the tetrathionate (S4I) pathway involving intermediates polythionates typical for acidophilic bacteria, and the branched thiosulfate oxidation pathway [6,28,39,61,62,63,64]. The main metabolic pathways and respective enzymes are shown in Figure 1.

The Sox multienzyme complex, first described using the model organism *Paracoccus panthotrophus*, is widespread in photo- and chemotrophic sulfur-oxidizing bacteria and appears to be the most prominent sulfur oxidation system. The metabolic pathway consists of at least two transcriptional units of 15 genes located on a 13kb DNA fragment. At least 7 genes are essential for complete sulfur oxidation with their protein products, namely SoxB, Sox(CD)_2_, SoxYZ, and SoxXA [41,62,65,66,67,68,69,70]. In addition to thiosulfate, the pathway employs the oxidation of sulfide, sulfur, tetrathionate, and sulfite. The transcription of the respective genes is induced by the presence of reduced sulfur compounds [26,71,72]. The central element of the Sox multienzyme pathway is the SoxXY protein complex. The sulfur of thiosulfate is covalently linked to the thiol of the active site SoxY-cysteine-sulfhydryl group of the SoxYZ by the SoxXA enzyme complex, from which the terminal sulfone group is subsequently hydrolyzed by the activity of the SoxB component. The sulfane sulfur of the residual SoxY-cysteine persulfide is oxidized by the Sox(CD)2 dehydrogenase complex, from which the sulfonate moiety is hydrolyzed by the activity of SoxB, thereby restoring SoxYZ, which completes the cycle [5,25,39,62,68,73,74]. The yield of the respective reaction is 8 electrons/1 mol of thiosulfate for the electron transport systems [62,75].

Reduced sulfur compounds serve as photosynthetic electron donors for two groups of phototrophic prokaryotes—green sulfur and purple sulfur bacteria [69]. As the model of photosynthetic sulfur oxidation *Allochromatium vinosum*, a member of *Chromatiaceae,* is used. All phototrophic members of the family *Chromatiaceae* use sulfide and elemental sulfur, and most of them also use thiosulfate as an electron donor, with sulfate as the end product [29]. The initial step of the branched thiosulfate oxidation pathway is carried out by the periplasmatic proteins SoxXA, SoxYZ, SoxB, and SoxL. Although they contain *sox* genes, there is an agreement that the *soxCD* genes that encode the essential Sox(CD)_2_ component of the reaction mechanism in *P. pantotrophus* are missing in their genomes. Therefore, sulfur intermediates as sulfur globules are formed. The globules produced via the thiosulfate oxidation are stored intracellularly by SgpA, SgpB, and SgpC proteins, forming an envelope [76]. The complex mechanism of oxidation of stored sulfur globules is used, with the activity of several enzymes of the *dsr* gene cluster, including reverse dissimilatory sulfite reductase (rDsr). The product of this reaction is sulfite, which is subsequently oxidized by the activity of the membrane-bound protein SoeABC (sulfite dehydrogenase) to sulfate, or by APS reductase and ATP sulfurylase. In addition to that, phototrophic sulfur-oxidizing bacteria oxidize sulfide using flavocytochrome c (FccAB) and sulfide:quinone oxidoreductases (SqrD and SqrF) [29,64,76,77,78].

Tetrathionate is a result of the oxidative metabolism of some sulfur-oxidizing bacteria, in which sulfur compounds are oxidized via the polythionate oxidation pathway. Tetrathionate is a stable sulfur compound under acidic conditions, distinct from other reduced sulfur compounds such as thiosulfate, sulfite, or sulfide. Thus, the polythionate oxidation pathway is typical for extremophilic representatives such as *Beta*- and *Gammaproteobacteria* representatives, i.e., genera Thiobacillus and *Acidithiobacillus* [63]. In this bacteria, thiosulfate is converted to tetrathionate by the activity of thiosulfate:quinone oxidoreductase or thiosulfate dehydrogenase (TsdA). Thiosulfate dehydrogenase catalyzes the formation of a sulfur–sulfur bond between two thiosulfate molecules, producing tetrathionate and yielding two electrons. The intermediate tetrathionate is subsequently oxidized by the tetrathionate hydrolase (TetH) [25,79,80,81,82,83,84] or coupled with glutathione with the activity of thiol dehydrotransferase (ThdT). It was described that in *Advenella kashmirensis*, the products are then oxidized by the actions of SoxB and SoxCD proteins and sulfite:acceptor oxidoreductase (SorAB) [85,86,87]. The basic oxidation and disproportionation sulfur reactions performed through the main metabolic pathways and enzymes of sulfur-oxidizing bacteria in energy gaining processes are shown in Figure 1.

Microorganisms oxidizing sulfur compounds are diverse in terms of metabolism. Some of them are able to use a variety of sulfur species as a primary energy source, whereas others prefer specific sulfur sources; moreover, some of them use sulfur oxidation as a supplementary energy source [88,89]. The great diversity of sulfur-oxidizing bacteria is reflected in the lack of a universal mechanism of sulfur oxidation in bacteria. Despite the interest of microbiologists, the genetic background with proteins involved in the oxidative processes of sulfur compounds is not yet fully understood and is complicated by the bidirectional activities of key enzymes. During the oxidation of sulfur compounds, several other enzymes are employed, such as reversed dissimilatory sulfite reductase (rDsr), sulfate adenylyltransferase (Sat), adenylyl-sulfate reductase (Apr), sulfite dehydrogenases (SorAB, SoeABC), sulfur dioxygenases (SdoA, SdoB) and heterodisulfide reductase (Hdr), persulfide dioxygenase (PDO) studied especially within genus (*Acidi*-) *Thiobacilus* [4,8,43,44,77,90,91,92,93,94,95,96], or a variety of their combinations. The physiology of some sulfur-oxidizing bacteria is waiting to be completely described; thus, we can assume that there are many undiscovered enzymes and enzymatic pathways [89]. Electrogenic sulfur oxidation (eSox) is one of the unique metabolic concepts described, in which different cells creating the same multicellular filament perform distinct redox half-reactions in anoxic and oxic environments. In oxygen-free sediment zones electrons are obtained and transported along the longitudinal axis of the filament to cells located near the sediment–water interface (where oxygen is available), and cells reduce oxygen and consume protons [97]. Our understanding of the sulfur cycle and the microorganisms involved in it has improved considerably in recent years. Yet, there are still significant questions regarding the biology of sulfur bacteria, factors that control the turnover of sulfur compounds, and the processes involved in the biogeochemical sulfur cycle.

### 2.4. Cold Sulfur Springs as a Source of Unexplored Life

The most widespread natural sulfur-rich environments are solfataras, which are geothermal sites in the vicinity of active volcanic environments [98]. Other examples of environments with the most marked sulfur cycling include hydrothermal vents, soda lakes, marine sediments, and acid mine drainage sites [32]. One of the inland environments harboring sulfur-oxidizing bacteria is the oxic–anoxic zones of lakes, where decaying organic matter serves as a source of reduced sulfur compounds [99]. Due to the high abundance of sulfur compounds in seawater, much of our knowledge about the sulfur cycle derives from marine sediments, which represent an important environment for sulfur bacteria. The distribution of sulfur microorganisms is not restricted to marine environments, and sulfur-oxidizing microorganisms also play important ecological roles in freshwaters [8]. However, our understanding of the microbiology of these environments is fragmented as the microbiota of freshwater environments are influenced, to various extents, by many physiochemical and biological factors [100,101,102]. Sulfur springs originate in subsurface water sources and are more stable for bacterial life compared to other environments. Cold springs are characterized by slowly changing parameters such as pH, temperature, dissolved gasses, and others, as water emerges from subsurface environments, and where oxygen and hydrogen sulfide exist in steep gradients [103,104]. The crucial factor influencing the presence and diversity of sulfur-oxidizing bacteria is the availability of reduced sulfur compounds, mainly sulfides in terrestrial sulfur waters. The hydrogen sulfide dissolved in freshwater may originate from either the emergence of sulfide emanations or bacterial sulfate reduction [105]. Natural springs with sulfide-rich water are common worldwide and are classified as areas with high microbial growth [106].

The microbial community structure of sulfur-oxidizing bacteria has been extensively studied, mostly in extreme hydrothermal environments [107,108]. In comparison, the knowledge of the sulfur-oxidizing bacteria living in cold sulfur freshwaters is limited. Sulfur springs are extremely diverse ecotones in terms of hydrogeology, temperature, hydrochemistry, and ecology, and support the growth of a variety of unique sulfur-oxidizing bacteria, including the colorless and phototrophic sulfur-oxidizing bacterial communities as a reflection of the aqueous chemistry of upwelling water [14,109,110]. The diversity may be observed also within the spring, where microhabitats with slightly different parameters are created [111]. Under such conditions, many of the sulfur metabolic pathways are simultaneously operating, with taxonomically various representatives of microbial populations engaged in the sulfur cycle using intermediates as electron donors or acceptors. Moreover, Vigneron et al. [8] stated that sulfur cycle intermediates may be key hubs for electron flow and energy production over a wide range of environmental conditions. Terrestrial mesophilic springs were shown to host rich microbial communities forming extensive white microbial mats [14,112,113]. Additionally, the cold sulfur spring in the non-geothermal area supported a distinctive bacterial community with a strings-of-pearls-like morphology, and the community was composed of bacterial/archaeal association [106,114,115].

Several cold sulfur springs have also been studied in arctic and permanently ice-covered ecosystems, and it was shown that the major energy generating processes were mediated by sulfur-oxidizing bacteria [116,117,118,119,120,121,122]. The utilization of sulfur compounds is a major energy gaining process supporting communities of cold sulfur springs in cave systems [112,123,124,125] and these springs are dominated by *Campylobacteria* (formerly *Epsilonproteobacteria*) and *Gammaproteobacteria*. From respective phyla, the most prominent genera observed at Frasassi and Acquasanta Terme cave systems (Italy) were *Sulfurovum* and *Sulfuricurvum* [125]. Similarly, terrestrial sulfidic springs represented a suitable environment for both *Campylobacteria* and *Gammaproteobacteria*, which could be considered core microbiota [104,110,112,126]. At the genus level, several genera are regularly observed and usually their presence correlates with the gradient of oxygen and sulfur-reduced compounds. Typical genera inhabiting sulfur springs in caves and terrestrial sulfur springs are shown in Figure 2.

In general, chemolithoautotrophic sulfur-oxidizing bacteria are localized within the sulfide–oxygen interface zones, competing with abiotic sulfide oxidation [1]. In cold sulfur spring environments at high sulfide and lower oxygen concentration zones, genera *Sulfurovum* and *Sulfuricurvum* tend to dominate [118]. Representatives of the genus *Thiothrix* are typical inhabitants of cold sulfur springs [105], however, contrary to *Sulfurovum* and *Sulfuricurvum* representatives, the genera *Thiothirix* and *Thiobacillus* are abundant in zones where the oxygen level is higher with the opposite gradient of hydrogen sulfide [123]. In addition, at least two species of the genus *Beggiatoa* are relatively prominent, especially two species within the biofilm [130]. *Beggiatoa*-related bacteria are found at steep oxygen and sulfide gradients, near the water–sediment interface, and usually flourish in microaerophilic environments [124,131,132]. Additionally, the microbiota of cold sulfur spring emanating from deep saline aquifers were dominated by unique chemolithoautotrophic sulfur-oxidizing *Thiomicrospira*-related bacterial species [117,133,134]. Similar evidence of the *Thiomicrospira*-related species was observed in a saline spring that emerged from the deep subsurface in Slovakia [127]. Interestingly, based on the relatively low 16S rRNA gene sequence similarities, the microbiota were composed of completely novel species of sulfur-oxidizing bacteria.

Cold sulfur springs were overlooked in microbiome analyzes for a long time, so the microbiota inhabiting this environment are largely unknown. From 11 isolates of cultivated sulfur-oxidizing bacteria from two cold sulfur springs in Slovakia, 6 showed 16S rRNA sequence similarities lower than 99% after the comparison with rRNA/ITS GenBank database. Moreover, 3 showed 16S rRNA sequence similarities lower than 97%, and single isolate showed similarity as low as 92.4% [128]. Similarly, 4 out of 13 DGGE bands in non-cultivation analysis of populations of SOBs in natural cold sulfur springs in Slovakia showed 16S rRNA similarities lower than 97%, in one case with similarity as low as 95% to monotypic bacterial genus *Thiofaba* [129]. Similarly, cultivation analysis of springs with high salinity showed that cultivable microbiota were mostly composed of novel, unidentified bacterial species, showing 16S rRNA gene sequence similarity from 97.55% to 98.47% [127].

Cold sulfur springs harbor unique, not yet explored bacterial communities [125,128,129] and are a neglected source of novel species of sulfur-oxidizing bacteria. Studies of cold sulfur springs, therefore, expand our knowledge of the bacteria involved in the biogeochemical sulfur cycle, their metabolism, ecology, and evolution, and may indicate the relationships between the sulfur microbial communities and the environment they inhabit.

## Figures and Tables

**Figure 1 microorganisms-11-01436-f001:**
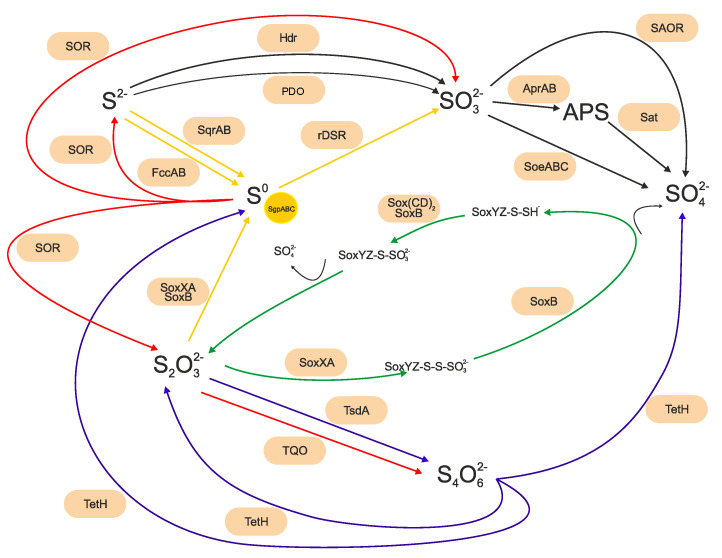
Simplified overview of the main sulfur oxidation pathways of reduced sulfur compounds used by sulfur-oxidizing bacteria. Core pathways are highlighted in colors, the red color indicates reactions observed in domain *Archaea*, the green indicates the pathway used by microorganisms with the whole Sox multienzyme complex, the yellow indicates reactions used by sulfur-oxidizing photosynthetic bacteria, and the reactions highlighted in purple are observed in acidophilic bacteria, especially in classes *Gamma*- and *Betaproteobacteria*. Reactions in black were observed in several different bacterial groups. For clarity, these are the main oxidation pathways, and many other reactions were observed in sulfur-oxidizing bacteria. Moreover, a variety of pathways could consist of their various combinations with the shown components connected by their reactants and products. SOR = sulfur oxygenase reductase, Hdr = heterodisulfide reductase, TQO = thiosulfate:quinone oxidoreductase, TsdA = thiosulfate dehydrogenase, TetH = tetrathionate hydrolase, SoxXA, SoxB, Sox(CD)_2_, SoxYZ = subunits of sox multienzyme complex, FccAB = flavocytochrome c, SqrAB = sulfide:quinone oxidoreductase, rDSR = reverse dissimilatory sulfite reductase, PDO = persulfide dioxygenase, SoeABC = sulfite dehydrogenase, AprAB = adenylyl-sulfate reductase, Sat = sulfate adenylyltransferase, SAOR = sulfite:acceptor dehydrogenase.

**Figure 2 microorganisms-11-01436-f002:**
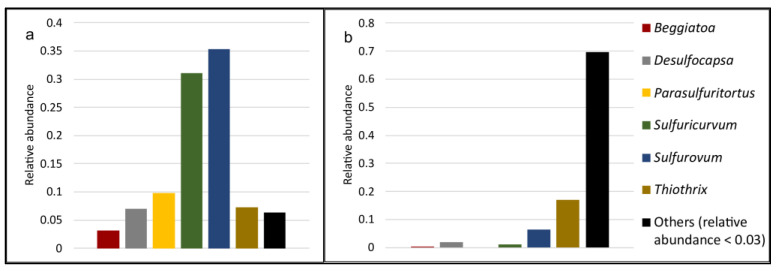
Relative abundance of dominant genera inhabiting sulfur springs in caves (**a**) and terrestrial sulfur springs (**b**). Sequences with a relative abundance lower than 0.03 were grouped as “Others”. Sequences obtained through listed studies [14,104,107,110,112,115,123,124,127,128,129] were downloaded and compared with the ITS/rRNA GenBank database. Results of identification were visualized as bar chart.

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
