# Peer review of "Cold Sulfur Springs—Neglected Niche for Autotrophic Sulfur-Oxidizing Bacteria"

_microorganisms, 2023, doi:10.3390/microorganisms11061436_

Round 1

Reviewer 1 Report

The author provides a review of the sulfur biological cycle and its new progress in cold sulfur springs,  a special environment. Overall, the content of this manuscript is suitable for publication in this journal. However, there are two issues that need to be addressed by the author.

1. In the current manuscript, most of the sulfur metabolism is focused on autotrophic bacteria, which must grow with sulfur as an energy source. However, the sulfur cycle in heterotrophic bacteria has been well appreciated in recent years, and the authors should recognize this field to make the manuscript more complete.

2. The oxidation of S0 (Fig.1), the authors focused on the anaerobic condition, while the aerobic oxidation by persulfide dioxygenase to sulfite is an important oxidation pathway, the authors should analyze the contribution of this pathway within the scope of the manuscript.

Reviewer 2 Report

The review "Cold Sulfur Springs - Neglected Niche for Sulfur-oxidizing Bacteria" is dedicated to the analysis of knowledge about Sulfur-oxidizing Bacteria living in cold springs. The authors analyzed a large papers and of published data, which examined various biochemical pathways of the sulfur cycle and microorganisms involved in this processes. The sulfur cycle is presented in the context of the evolution of the Earth, the role of various factors in the evolutionary radiation of microorganisms of this group, as well as genes responsible for the synthesis of enzymes that ensure the transformation of various sulfur-containing substrates. The problem under discussion is very interesting and multifaceted.

As a rule, reviews should present not only a selection of references about a particular process, but also contain illustrative material that makes it possible to understand the difference between processes, evaluate the transformations of substrates and its function in the life of individual participants in the sulfur cycle. The article practically does not contain illustrations, descriptions of reactions and looks very boring and uninformative and is unlikely to be in demand by other researchers. Given the large volume of scientific publications on various groups of microorganisms involved in the sulfur cycle, it will be much more efficient to use other publications, rather than the presented review.

It would be much more interesting to present the processes of the sulfur cycle in different biotopes and show differences in the diversity and metabolic pathways of bacteria and archaea depending on salinity, temperature, and other factors. Such a presentation of the material will correspond to the title of the paper, now there is a discrepancy between the content and the title.

It would be good to present data on physicochemical parameters in various sources in the form of a table, where the diversity of dominant taxa and differences in the structure of microbial communities are shown, indicating the study area. The given data on Cold sulfur springs as a source of unexplored life are not convincing and require more justification.

Round 2

Reviewer 1 Report

I have no further comments. 

Reviewer 2 Report

Too bad you didn't illustrate your paper.